Long-term oncologic outcomes of laparoscopic nephroureterectomy versus open nephroureterectomy for upper tract urothelial carcinoma: a systematic review and meta-analysis

Zhang Su
Luo You
Wang Cheng
Fu Sheng-Jun
Yang Li professoryangli@163.com
Department of Urology, Lanzhou University Second Hospital , Lan Zhou , China
Huisman Henkjan
Electronic publication date: 2016 May 31
Publication date: 2016
Volume: 4
Electronic Location ID: e2063
Received 2016 Feb 20; Accepted 2016 May 1
Copyright: ©2016 Zhang et al.
Copyright year: 2016
Copyright holder: Zhang et al.
License: This is an open access article distributed under the terms of the Creative Commons Attribution License, which permits unrestricted use, distribution, reproduction and adaptation in any medium and for any purpose provided that it is properly attributed. For attribution, the original author(s), title, publication source (PeerJ) and either DOI or URL of the article must be cited.
License URL: https://creativecommons.org/licenses/by/4.0/

Keywords: Upper tract urothelial carcinoma, Laparoscopic nephroureterectomy, Open nephroureterectomy, Meta-analysis

Funding: Science and Technology Board of Chengguan District, Lanzhou 2015 CGKJ217 This work was founded by Science and Technology Board of Chengguan District, Lanzhou (2015 CGKJ217). The funders had no role in study design, data collection and analysis, decision to publish, or preparation of the manuscript.

==============================
Background. Several factors have been validated as predictors of disease recurrence in upper tract urothelial carcinoma. However, the oncological outcomes between different surgical approaches (open nephroureterectomy versus laparoscopic nephroureterectomy, ONU vs LNU) remain controversial. Therefore, we performed a meta-analysis to evaluate the oncological outcomes associated with different surgical approaches.

Methods. We conducted an electronic search of the PubMed, Embase, ISI Web of Knowledge and Cochrane Library electronic databases through November 2015, screened the retrieved references, collected and evaluated the relevant information. We extracted and synthesized the corresponding hazard ratios (HRs) and 95% confidence intervals (95% CI) using Stata 13.

Results. Twenty-one observational studies were eligible for inclusion in the meta-analysis. The results of the meta-analysis showed no differences in the intravesical recurrence-free survival (IRFS), unspecified recurrence-free survival (UnRFS) and overall survival (OS) between LNUandONU. However, improvements in the extravesical recurrence free survival (ExRFS) and cancer specific survival (CSS) were observed inLNU. The pooled hazard ratios were 1.05 (95% CI [0.92–1.18]) for IRFS, 0.80 (95% CI [0.64–0.96]) for ExRFS, 1.10 (95% CI [0.93–1.28]) for UnRFS, 0.91 (95% CI [0.66–1.17]) for OS and 0.79 (95% CI [0.68–0.91]) for CSS.

Conclusion. Based on current evidence, LNU could provide equivalent prognostic effects for upper tract urothelial carcinoma, and had better oncological control of ExRFS and CSS compared to ONU. However, considering all eligible studies with the intrinsic bias of retrospective study design, the results should be interpreted with caution. Prospective randomized trials are needed to verify these results.

Introduction

Upper tract urothelial carcinoma (UTUC), accounting for only 5% of all urothelial cancers, is a rare malignancy with high risk for disease recurrence and mortality (Roupret et al., 2015). Given its high potential for recurrence and poor prognosis, assessment of the predictive factors appears to be increasingly significant. Tumor multifocality, previous bladder cancer and concomitant carcinoma in situ (CIS) have been validated as predictors of intravesical recurrence, which is also associated with different surgical approaches (Xylinas et al., 2013; Xylinas et al., 2014). The standard treatment for UTUC is nephroureterectomy with bladder cuff excision. The treatment approaches include open nephroureterectomy (ONU) and laparoscopic nephroureterectomy (LNU). Compared with ONU, a traditional approach which has durable oncologic control, LNU has shown several advantages with fewer adverse intra- and perioperative outcomes as a minimally invasive treatment since being first introduced by Clayman in 1991 (Clayman et al., 1991; Simone et al., 2009). However, compared to ONU, whether LNU has equivalent oncological outcomes, such as cancer-specific survival (CSS) and intravesical recurrence-free survival (IRFS), remains controversial (Kim et al., 2015; Xylinas et al., 2014). We aimed to perform a meta-analysis to evaluate the oncological control associated with different surgical approaches (ONU vs LNU).

Methods

Search and screen strategy

A systematic literature search of Embase, PubMed, ISI Web of Knowledge and Cochrane Library was conducted to retrieve UTUC studies comprising both surgical approaches (ONU and LNU) through November 1, 2015. The search key words included open nephroureterectomy, laparoscopic nephroureterectomy, upper tract urothelial carcinoma, and others. The detailed search strategy is presented in Supplemental Information 2. We also screened the citations in the retrieved articles for any relevant studies. Two independent investigators (S Zhang and Y Luo) conducted the initial screening by reviewing the title and abstract. Then, the full-text articles satisfying the inclusion criteria were reviewed. Clinical studies recording any evaluation of the surgical approach on oncological outcomes, including intravesical recurrence-free survival (IRFS), extravesical recurrence-free survival (ExRFS), unspecified recurrence-free survival (UnRFS, reported as disease recurrence but not explicitly defined as IRFS or ExRFS), cancer-specific survival (CSS) or overall survival (OS), were eligible. Articles were excluded if they met any of the following criteria: (1) the aforementioned outcomes were not described; (2) patients were treated by hand assisted laparoscopic nephroureterectomy; or (3) studies included overlapping patients or duplicated data. Instead, the study with the largest sample size would be selected if more than one study included overlapping patients. This systematic review was performed according to the Preferred Reporting Items for Systematic Reviews and Meta-Analyses statement (PRISMA) (Moher et al., 2010). Ethical approval and patient consent were waived because all available data were extracted from previous publications.

Data extraction and assessment of methodological quality

The basic information including first author, year of publication, region, recruitment period, number of patients who underwent LNU or ONU, age of patients, follow-up, oncological outcomes, and adjusted factors were extracted by two researchers (S Zhang and Y Luo) independently. Any disagreement or uncertainty was determined by group discussion, and a consensus was obtained. The data were extracted from the original articles. For incomplete data, we attempted to contact the corresponding author to acquire primary data. During data extraction, multivariate outcomes rather than univariate outcomes were preferred when both results were provided. If multivariate results were not available, univariate outcomes were an alternative to conduct this analysis. Publication bias and sensitivity analyses were applied. The quality assessments of cohort studies were conducted according to the Newcastle-Ottawa Scale (NOS), which was developed to assess bias risk including three domains with eight items. Five or more stars out of a total of nine stars was regarded as good quality (Wells et al., 2008).

Statistical analysis

All data and analysis were conducted using STATA 13 software (Stata Corp LP, College Station, TX, USA). The survival outcomes were evaluated by hazard ratios (HR) and 95% confident intervals. I2 statistics and the chi-square test were calculated for heterogeneity detection. When P ≥ 0.1 and I2 ≤ 50%, a fixed-effects model was performed; otherwise, a random effects model was applied. An inverse variance method was used to calculate the pooled hazard ratio. Sensitivity analyses were conducted to test the stability of the pooled results. Egger’s test for publication bias was performed only in outcomes that enrolled more than ten studies (Egger et al., 1997). Additionally, we conducted subgroup and multivariable meta-regression in IRFS according to the different approach of LNU (retroperitoneal vs. transperitoneal), sample size of LNU (<100 vs ≥100) and publication year. A P value of less than 0.05 was deemed statistically significant.

Results

Description of included studies

In total, 1,506 citations were retrieved by the initial search strategy. After three rounds of screening, there were 21 cohort studies for quantitative synthesis. The PRISMA flow diagram is presented in Fig. 1. Table 1 shows the detailed characteristics of the included studies. The Newcastle Ottawa Scale (NOS) assessment showed that all included cohort studies had relatively well controlled quality.

Figure 1 Screening flow diagram.

Table 1 Characteristics of the studies included in the systematic review.

Study	Country	Duration	N of pts (ONU/LNU)	Age (yrs) (ORNU/LRNU)	Follow up (month) (ONU/ LNU)	Outcomes	Approach of LNU	NOS	Adjusted factors	
Favaretto et al. (2010)	USA	2002–2008	109/53	Md73 (IQR67-78) Md71 (IQR64-76)	Md23	UnRFS, IRFS	Mixed	6	Age, ASA, pT, Grade, pN, CIS, PBC	
Fradet et al. (2014)	Canada	1990.1–2010.6	267/345	Md67 (IQR59-75)	Md24.8 (IQR7.69-56.76)	IRFS	NA	7	Age, Smoking, PH, Previous abdominal radiotherapy, DUM, CIS, TL, AC	
Ito et al. (2013)	Japan	2005.12–2008.11	39/33	NA	R2.6-39.3	IRFS	RE	6	Sex, Age, TS, pT, UC, Grade, CIS, Histology type, AC	
Kim et al. (2015)	Korea	1992–2012	271/100	Md64.7 (IQR57.7-70.8)	Md50.8 (IQR26.6-103.6)	OS, CSS, IRFS	TR	8	Age, ASA, PBC, UC, pT,Grade, LVI, Variant histology of urothelial carcinoma, TL, SM, AC	
Kitamura et al. (2014)	Japan	1995.4–2010.8	34/65	Md69 (R32-88) Md65 (R53-71)	Md70 (R6-192)	IRFS	Mixed	6	Grade	
Zou et al. (2014)	CHN	1999.1–2013.2	101/21	Mn63.7 (R35-80)	Md53 (R3-159)	IRFS, CSS	TR	7	Gender, PH, TS, TL, Size, Appearance, Necrosis, pT, Grade, Multifocality, CIS, SM, LVI	
Yafi et al. (2012)	Canada	1990-/	591/46	Md68 (IQR61-75)	Md37 (IQR18-68)	ExRFS,CSS	NA	6	Age, Race, Gender, TL, pT, Grade, CIS, LVI, pN	
Walton et al. (2011)	Multi central	1987–2008	703/70	Md68 (IQR61-75)	Md34 (IQR15-65)	ExRFS, CSS	NA	7	Age, Gender, Race, PBC, DUM, TL, Grade, pT, pN, LVI, CIS	
Taweemonkongsap et al. (2008)	Thailand	2001.4–2007.1	29/31	Mn66.8 (R39-88) Mn63.8 (R26-79)	Mn27.9 (R3-63) Mn26.4 (R3-72)	UnRFS	RE	5	pT, Grade	
Metcalfe et al. (2012)	Canada	1990–2010	403/446	Mn69.7 (SD10.7)	Mn26.4 (R7.2-60)	UnRFS, OS	NA	8	Region, Age, Symptoms, TL, pT, Grade, CIS, PBC, NeoAC, AC, Salvage chemotherapy, Salvage radiation therapy, SM, Smoking, Previous abdominal RT, pN	
Kume et al. (2006)	Japan	1996–2003	28/13	Mn65.07 (SD9.46) Mn65.31 (SD10.69)	Mn55.7 (SD29.4) Mn34.2 (SD10.9)	IRFS	RE	6	Multiple tumors, pT, Grade, OT	
Koda et al. (2007)	Japan	1995.1–2005.8	27/29	Mn67.4 (SD11.3) Mn71.4 (SD8.2)	Mn46.2 (R1-97) Mn16.4 (R1-57.5)	IRFS	RE	6	Sex, Side, Age, pT, Grade, OT, AC, PBC	
Ploussard et al. (2015)	Multicentres	1989–2012	2826/922	Md70 (IQR60-74)	Md32.7 (IQR13.6-67.4)	IRFS, CSS	NA	7	Age, Sex, Ureter location, Multifocality, LN, DUM, pT, High grade, CIS, AC	
Rieken et al. (2014)	Muticentres	1987–2007	2042/450	Md69.2 (IQR62-77)	Md36	ExRFS, OS, CSS	NA	5	Univariable Cox regression	
Fairey et al. (2013)	Canada	1994–2009	403/446	Md70.5 Md72.4	Md26.4 (IQR7.2-60)	UnRFS, OS, CSS	Mixed	7	Age, Sex, AC, pT, pN, Grade, SM	
Capitanio et al. (2009)	Muticentres	1987-2007	979/270	Mn68.3 (R27-97) Mn70.2 (R36-94)	Md49	ExRFS, CSS	NA	7	Age, pT, pN, Grade, LVI, ECOG PS, pN, PBC, Previous endoscopy, CIS	
Ariane et al. (2012)	France	1995–2010	459/150	Md69.8 (R60.9-76) Md69.5 (R63-77)	Md27 (R10-48)	UnRFS, CSS,	TR	6	Gender, Age, ASA physical status, TL, pT, Grade, pN, LVI	
Kobayashi et al. (2012)	Japan	2005.1–2009.4	151/137	Md71.4 (R32-89)	Md20.2 (R3.0-61.6)	IRFS	RE	6	TL, Time of ligation of the ureter, UC.	
Terekawa et al. (2008)	Japan	2000.1–2005.12	111/66	Mn71.3 (SD9.6) Mn 68.7 (SD9.5)	Mn31 (R12.0-80.5)	IRFS	RE	7	Age, TS, TL, Multifocality, OT, DUM, pT, Grade, pN, LVI, SM	
Ishikawa et al. (2010)	Japan	1990–2005	165/43	Md70 (R39-90)	Md8 (R2-105)	IRFS, CSS	RE	5	Univariable Cox regression	
Xylinas et al. (2013)	France	1995–2009	350/132	Mn69.2 (IQR60-76)	Mn39.5 (IQR25-60)	IRFS	NA	6	Age, Gender, TL, Multifocality, PBC, Endoscopic management, pT, Grade, CIS, LVI, pN	
Notes.

List of abbreviations yrs years

N of pts number of patients

Mn mean

Md median

R range

IQR interquartile range

NA not applicable

IRFS intravesical recurrence free survival

ExRFS extravesical recurrence free survival

UnRFS unspecified recurrence free survival

OS overall survival

CSS cancer specific survival

RE retroperitoneal

TR transperitoneal

NOS Newcastle-Ottawa Scale

pT pTstage

pN pNstage

TL tumor location

PH previous hydronephrosis

DUM distal ureter management

TS tumor side

UC urinary cytology

SM surgical margin

PBC previous bladder cancer

ASA American Society of Anesthesiology physical status

AC adjuvant chemotherapy

OT operation time

CIS carcinoma in situ

LVI lymphovascular invasion

ECOG PS Eastern Cooperative Oncology Group performance score

Survival outcomes

Oncological recurrence

The IRFS was reported in thirteen articles, which included LNU (n = 1,959) and ONU (n = 4,281) (Favaretto et al., 2010; Fradet et al., 2014; Ishikawa et al., 2010; Ito et al., 2013; Kim et al., 2015; Kitamura et al., 2014; Kobayashi et al., 2012; Koda et al., 2007; Kume et al., 2006; Ploussard et al., 2015; Terakawa et al., 2008; Xylinas et al., 2013; Zou et al., 2014). The meta-analysis results showed no significant difference in the IRFS between LNU and ONU management (HR 1.05, 95% CI [0.92–1.18]; P = 0.134, I2 = 31.1%; Fig. 2). The ExRFS was described in four studies including patients who underwent LNU (n = 836) and ONU (n = 4,315) (Capitanio et al., 2009; Rieken et al., 2014; Walton et al., 2011; Yafi et al., 2012). The pooled results showed that LNU management decreased the risk of extravesical recurrence (HR 0.80, 95% CI [0.64–0.96]; P = 0.859, I2 = 0.0%; Fig. 3). Five studies including LNU (n = 1,126) and ONU (n = 1,403) (Ariane et al., 2012; Fairey et al., 2013; Favaretto et al., 2010; Metcalfe et al., 2012; Taweemonkongsap et al., 2008) reported the UnRFS. The pooled analysis of the available HRs showed that the different surgical procedures were not significantly correlated with disease recurrence (HR 1.10, 95% CI [0.93–1.28]; P = 0.337, I2 = 12.0%; Fig. 4).

Figure 2 Forest plot of Intravesical Recurrence Free Survival (IRFS) hazard ratio.

Figure 3 Forest plot of Extravesical Recurrence Free Survival (ExRFS) hazard ratio.

Figure 4 Forest plot of Unspecified Recurrence Free Survival (UnRFS) hazard ratio.

Mortality

Among the four studies that provided the HRs of OS, there were 1,442 LNU patients and 3,119 ONU patients (Fairey et al., 2013; Kim et al., 2015; Metcalfe et al., 2012; Rieken et al., 2014). There was significant heterogeneity (P = 0.091, I2 = 53.7%; Fig. 5), and a random model was applied. The model showed that neither LNU nor ONU significantly increased the risk in the overall survival (HR 0.91, 95% CI [0.66–1.17]; Fig. 5). The CSS was described in ten articles, in which 2,518 patients were treated by LNU and 8,342 patients were treated by ONU (Ariane et al., 2012; Capitanio et al., 2009; Fairey et al., 2013; Ishikawa et al., 2010; Kim et al., 2015; Ploussard et al., 2015; Rieken et al., 2014; Walton et al., 2011; Yafi et al., 2012; Zou et al., 2014). The pooled results indicated that LNU could improve the cancer specific survival (HR 0.79, 95% CI [0.68–0.91]; P = 0.186, I2 = 28.1%; Fig. 6).

Figure 5 Forest plot of Overall Survival (OS) hazard ratio.

Figure 6 Forest plot of Cancer Specific Survival (CSS) hazard ratio.

Subgroup analysis and multivariable meta-regression for IRFS

In subgroup analysis for the effect of different approaches of LNU on IRFS, no difference were seen among people with retroperitoneal laparoscopy (HR 1.04, 95% CI [0.77–1.32]; P = 0.598, I2 = 0.0%; Fig. 7) and transperitoneal laparoscopy (HR 0.81, 95% CI [0.48–1.13]; P = 0.548, I2 = 0.0%; Fig. 7). The subgroup of five studies with sample sizes of LNU more than 100 had a combined HR of 1.31 (95% CI[0.92–1.70]) with significant heterogeneity (P = 0.011, I2 = 69.5%; Fig. 8), while the subgroup of eight studies with sample sizes of less than 100 had a combined HR of 0.97 (95% CI [0.67–1.16]) without significant heterogeneity (P = 0.919, I2 = 0.0%; Fig. 8). Multivariable meta-regression showed no particular influence of different approaches of LNU (P = 0.431), sample size (P = 0.899) and publication year (P = 0.729) on the results.

Figure 7 Forest plot of subgroup analysis for IRFS - stratified by LNU approach.

Figure 8 Forest plot of subgroup analysis for IRFS—stratified by sample size.

Publication bias and sensitivity analysis

The publication bias detection was conducted by Egger’s asymmetric test and only for IRFS outcomes. The P value of the linear regression was 0.515, and no significant publication bias was observed (Fig. S1). We also performed sensitivity analyses of IRFS and CSS, and no significance change was observed (Figs. S2 and S3).

Discussion

Recently, some retrospective studies have paid attention to the predictors of disease recurrence in patients with UTUC after RNU; these studies found that CIS, previous bladder cancer, laparoscopic surgery and distal ureteral management were risk factors for disease recurrence (Xylinas et al., 2014). A systemic review concluded that a laparoscopic approach significantly increased the risk of intravesical recurrence (Seisen et al., 2015). However, there are many studies that suggested that laparoscopic surgery could provide equivalent oncologic control compared with open surgery (Favaretto et al., 2010; Ishikawa et al., 2010). Therefore, we conducted this meta-analysis enrolling twenty-one retrospective studies that contained various oncologic outcomes to assess whether LNU would show a tendency toward a poor prognosis for UTUC patients.

Open nephroureterectomy, the traditional surgical approach that could support a durable tumor control, has long been accepted as the standard surgical treatment for UTUC, especially for high-risk UTUC (Roupret et al., 2015). As a viable minimally invasive therapy, LNU was developed in an effort to reduce the morbidity of the surgical management and had advantages of lesser blood loss, shorter hospital stay and oncologic outcomes compared with ONU. With a median follow-up of 45 months of 1,261 UTUC patients who underwent ONU (n = 926) or LNU (n = 335), Xylinas et al., 2014 showed that the laparoscopic approach was associated with a higher risk of intravesical recurrence compared with open surgery (HR = 1.5, 95% CI [1.17–1.93]). It was suggested that the high pressure pneumoperitoneum during LNU might trigger tumor dissemination and could result in a higher rate of recurrence, which contributed to the debate on oncologic outcomes of UTUC patients after laparoscopic procedures. Few cases of laparoscopic port-side seeding were reported in some literature in the early years, and Roupret et al. (2015) proposed that ensuring a closed system during laparoscopic surgery and avoiding direct contact between instruments and tumors might favor tumor control. In 150 laparoscopic surgeries, (Ariane et al., 2012) reported three cases of laparoscopic port-side seeding occurred in early experiences. After the widespread use of laparoscopic bags for specimen extraction, no cases happened. Our pooled results demonstrated that LNU could provide equivalent tumor control of intravesical recurrence and unspecified RFS compared to ONU. However, the majority of the enrolled studies reported negative control for ONU in extravesical recurrence. Our results showed that ONU was an independent risk for ExRFS. However, considering that there were only four articles enrolled, whether open surgery increased the risk of extravesical recurrence still needs further exploration. Our analysis based on the current evidence did not support the view that laparoscopic surgery increased the risk of disease recurrence of patients with UTUC after nephroureterectomy.

Regarding survival outcomes, our data demonstrated that LNU was comparable to ONU in overall survival, and superior in cancer specific survival. In the first randomized prospective study, the cancer specific survival rate and the metastasis free survival rate were significantly different between the LNU and ONU groups, which favored ONU after matching for pT3 and high-grade tumors (Simone et al., 2009). In the 2015 EAU guidelines, invasive or large (T3/T4 and/or N+/M+) tumors were deemed as contraindications for a laparoscopic approach (Roupret et al., 2015). Recently, Kim et al. (2015) retrospectively analyzed the data of 371 UTUC patients who underwent ONU (n = 271) or LNU (n = 100); the results indicated that LNU had worse five-year OS and CSS rates than the ONU group only in locally advanced disease (pT3/T4) after stratifying by pathological stages. However, this conclusion was not identified in Arian’s research in tumors of the pathological stages of pT3/T4 (Ariane et al., 2012). A recent study including 749 muscle-invasive UTUC patients who underwent ONU (n = 527) or LNU (n = 222) also indicated that the oncological outcomes of LNU were not inferior to the outcomes of ONU (Miyazaki et al., 2016). Although our analysis did not include a subgroup analysis of survival outcomes (OS and CSS) in locally advanced UTUC patients because of a lack more relevant survival data, our results might be reliable because the majority of our data were extracted from multivariate analyses, the majority of which adjusted for the effect of tumor stages and grades. Furthermore, a previous systematic review also showed that no significant differences in the stages of pT3/T4 or pathologic grades were observed in the LNU group compared with the ONU group (Ni et al., 2012).

Previous systematic reviews on the oncologic outcomes comparing LNU with ONU were published in 2012 (Ni et al., 2012; Rai et al., 2012). These cumulative analyses conducted by using non-time to event data suggested that LNU could offer reliable perioperative safety and comparable oncologic efficacy compared to ONU. The meta-analysis conducted by Ni et al. indicated that LNU could improve the 5-yr CSS and decrease the rates of the overall recurrence and bladder recurrence. Recently, Seisen et al. (2015) observed that LNU was a significant predictor of the IRFS in their meta-analysis enrolling in six studies (HR 1.62; 95% CI [1.18–2.22]). After enrolling more available HRs extracted from multivariate or univariable Cox regression, our results showed there was no significant difference in IRFS between LNU and ONU management, different from the studies of Ni et al. and Seisenet al. We thought the conclusion that LNU could improve the CSS should be interpreted cautiously, although this conclusion was consistent with the previous opinion of Ni et al.

Several limitations must be acknowledged in our meta-analysis. First, this meta-analysis was based on retrospective studies. Although all of these studies were of high quality (>5 stars) according to the modified Newcastle-Ottawa Scale, the intrinsic bias of cohort studies existed. Next, the covariates controlled in the Cox regression analysis were different, which might introduce bias into our analysis. Finally, the time interval of the studies enrolled was more than 20 years. During this period, improvements in surgical techniques and medical materials should be considered. Additionally, these analyses did not include hand assisted laparoscopic nephroureterectomy because its relevant HR by Cox regression analysis for oncologic outcomes was reported in few literatures, and it is frequently deemed as an inferior approach compared with LNU or ONU in terms of the recurrence free survival and intravesical recurrence free survival rates (Kitamura et al., 2014). Given the low incidence of UTUC, high-quality level data were so scarce that our results should be interpreted cautiously. The oncological outcomes of LNU and ONU should be verified by prospective randomized controlled trials, especially for locally advanced disease.

Conclusion

Based on our meta-analysis of the current evidence, LNU could provide equivalent prognostic effects for upper tract urothelial carcinoma as ONU, and LNU had better results in the ExRFS and CSS. However, considering all eligible studies with the intrinsic bias of retrospective study design, the results should be interpreted with caution, and prospective randomized controlled trials are still needed.

Supplemental Information

Figure S1 Funnel plot to assess the risk of publication bias

Click here for additional data file.

Figure S2 Sensitivity analysis of IRFS

Click here for additional data file.

Figure S3 Sensitivity analysis of CSS

Click here for additional data file.

Supplemental Information 1 PRISMA_2009_checklist

Details of the PRISMA 2009 Checklist

Click here for additional data file.

Supplemental Information 2 Detailed search strategy for PubMed

Click here for additional data file.

Data S1 Raw data of eligible studies

Click here for additional data file.

Additional Information and Declarations

Competing Interests

Author Contributions

Data Availability

The authors declare there are no competing interests.

Su Zhang conceived and designed the experiments, performed the experiments, analyzed the data, contributed reagents/materials/analysis tools, wrote the paper.

You Luo conceived and designed the experiments, performed the experiments, contributed reagents/materials/analysis tools.

Cheng Wang analyzed the data, contributed reagents/materials/analysis tools, prepared figures and/or tables.

Sheng-Jun Fu prepared figures and/or tables.

Li Yang reviewed drafts of the paper.

The following information was supplied regarding data availability:

The raw data has been supplied as Data S1.

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
