# Peer review of "Long-term oncologic outcomes of laparoscopic nephroureterectomy versus open nephroureterectomy for upper tract urothelial carcinoma: a systematic review and meta-analysis"

_PeerJ, doi:10.7717/peerj.2063_

## Round 0.1 · original submission · Major Revisions

· Academic Editor

Major Revisions

Three reviewers have given their fair responses to this well-written paper. There is some debate as to what this paper adds to the 2012 EU paper. Please clarify the differences between the similar EU 2012 paper in your revision.

Reviewer 1 ·

Basic reporting

The systematic review performed is similar to the one from Shaobin Ni et al, Eur Urol 2012. The major difference is the use of more recent articles. However, the design of this systematic review is good and the analysis is well performed.

Major conclusions in the manuscript are:
1. No differences between LNU and ONU for intravesical recurrence-free survival, unspecified recurrence-free survival and overall survival
2. The extravesical recurrence free survival and cancer specific survival is better in LNU versus ONU.

I think the last conclusions have to be taken with a lot of cautious.
First, the writers mention that ONU was an independent risk for extravesical recurrence free survival. However, considering that there were only four articles enrolled, this risk of extravesical recurrence still needs further exploration. I think that’s a good description.
However, biggest problem with including restrospective studies is selection bias. Patient with known extensive disease (T3b or T4) most likely undergo an ONU. The number of patients with these kind of massive disease who undergo deliberately a LNU will be extremely limited. So the comparison between these groups is very difficult.
To my opinion this article is well written and underlines that LNU is feasible and safe, but the conclusion that the CSS improves due to the technique used is not right. I think these conclusion have to be weakened.

Experimental design

The design of this systematic review is good and the analysis is well performed.

Validity of the findings

The data are robuust, but conclusion should be taken with cautious since the data used were extracted from retrospective studies.

Additional comments

The systematic review performed is similar to the one from Shaobin Ni et al, Eur Urol 2012. The major difference is the use of more recent articles. However, the design of this systematic review is good and the analysis is well performed.

Major conclusions in the manuscript are:
1. No differences between LNU and ONU for intravesical recurrence-free survival, unspecified recurrence-free survival and overall survival
2. The extravesical recurrence free survival and cancer specific survival is better in LNU versus ONU.

I think the last conclusions have to be taken with a lot of cautious.
First, the writers mention that ONU was an independent risk for extravesical recurrence free survival. However, considering that there were only four articles enrolled, this risk of extravesical recurrence still needs further exploration. I think that’s a good description.
However, biggest problem with including restrospective studies is selection bias. Patient with known extensive disease (T3b or T4) most likely undergo an ONU. The number of patients with these kind of massive disease who undergo deliberately a LNU will be extremely limited. So the comparison between these groups is very difficult.
To my opinion this article is well written and underlines that LNU is feasible and safe, but the conclusion that the CSS improves due to the technique used is not right. I think these conclusion have to be weakened.

Reviewer 2 ·

Basic reporting

Clear paper. A fine description of the performed study to evaluate the oncological outcomes of open nephroureterectomy and laparscopic nephroureterectomy.

Experimental design

Meets the PRISMA guidelines. Analysis are well performed. Please mention the outcome of the I2 and the Chi-square statistic in the results.

Validity of the findings

Used statistics are sufficient.

Additional comments

Some typos, for example figure 1.

Reviewer 3 ·

Basic reporting

Actually, in 2012, a similar paper has been published in EUROPEAN UROLOGY (Ni S, Tao W, Chen Q, Liu L, Jiang H, Hu H, Han R, Wang C. Laparoscopic versus open nephroureterectomy for the treatment of upper urinary tract urothelial carcinoma: a systematic review and cumulative analysis of comparative studies. Eur Urol. 2012 Jun;61(6):1142-53.) In this paper, a total of 21 eligible studies were identified, and conducted similar conclusions as yours. So I cannot get the meaning why you conducted this meta-analysis.

Experimental design

In this study, the subgroup analysis based on 1) publication year or study duration, 2) LNU approach, 3) sample size must be conducted, as they are important confounding factors.

Validity of the findings

Please provide the results of publication bias and sensitivity analysis as supplementary information. The term “results were omitted” is NOT enough.

Additional comments

Dr. Zhang and colleagues conducted this systematic review and meta-analysis by pooling all eligible studies, with aim to oncological outcomes between open nephroureterectomy (ONU) and laparoscopic nephroureterectomy (LNU). The efforts are encouraging; however, it cannot be accepted for publication as it current stands.

---

## Round 0.2 · accepted · Accept

· Academic Editor

Accept

The manuscript has been adapted based on the comments of the reviewers. I agree with the reviewers that your adaptations make the manuscript acceptable.

Reviewer 1 ·

Basic reporting

With current changes I think this manuscript can be accepted for publication.
I have no additional comments.

Experimental design

No new comments

Validity of the findings

No new comments

Additional comments

No new comments

Reviewer 3 ·

Basic reporting

No comments

Experimental design

No comments

Validity of the findings

No comments

Additional comments

The comments raised have been addressed carefully.